# Bioinformatic Reconstruction and Analysis of Gene Networks Related to Glucose Variability in Diabetes and Its Complications

**DOI:** 10.3390/ijms21228691

**Published:** 2020-11-18

**Authors:** Olga V. Saik, Vadim V. Klimontov

**Affiliations:** 1Laboratory of Endocrinology, Research Institute of Clinical and Experimental Lymphology—Branch of the Institute of Cytology and Genetics, Siberian Branch of Russian Academy of Sciences (RICEL—Branch of IC&G SB RAS), 630060 Novosibirsk, Russia; klimontov@mail.ru; 2Laboratory of Computer Proteomics, Federal Research Center Institute of Cytology and Genetics, Siberian Branch of the Russian Academy of Sciences (ICG SB RAS), 630090 Novosibirsk, Russia

**Keywords:** diabetes, glucose variability, hyperglycemia, hypoglycemia, complications of diabetes, gene networks, ANDSystem, gene prioritization

## Abstract

Glucose variability (GV) has been recognized recently as a promoter of complications and therapeutic targets in diabetes. The aim of this study was to reconstruct and analyze gene networks related to GV in diabetes and its complications. For network analysis, we used the ANDSystem that provides automatic network reconstruction and analysis based on text mining. The network of GV consisted of 37 genes/proteins associated with both hyperglycemia and hypoglycemia. Cardiovascular system, pancreas, adipose and muscle tissues, gastrointestinal tract, and kidney were recognized as the loci with the highest expression of GV-related genes. According to Gene Ontology enrichment analysis, these genes are associated with insulin secretion, glucose metabolism, glycogen biosynthesis, gluconeogenesis, MAPK and JAK-STAT cascades, protein kinase B signaling, cell proliferation, nitric oxide biosynthesis, etc. GV-related genes were found to occupy central positions in the networks of diabetes complications (cardiovascular disease, diabetic nephropathy, retinopathy, and neuropathy) and were associated with response to hypoxia. Gene prioritization analysis identified new gene candidates (*THBS1, FN1, HSP90AA1, EGFR, MAPK1, STAT3, TP53, EGF, GSK3B*, and *PTEN*) potentially involved in GV. The results expand the understanding of the molecular mechanisms of the GV phenomenon in diabetes and provide molecular markers and therapeutic targets for future research.

## 1. Introduction

The concept of blood glucose variability (GV) attracts increasing attention in both basic and clinical science. Maintenance of GV within normal values is provided by a complex regulatory molecular-genetic network, which remains not fully understood. Many patients with diabetes have increased GV, which might be a major therapeutic challenge [1,2]. The highest GV is observed in insulin-treated type 1 and type 2 diabetic subjects. However, the initial increase in GV can be revealed in persons with impaired glucose tolerance [3].

A growing body of evidences indicates a role of GV in development of diabetic vascular complications and clinically important outcomes. In the observational Finnish Diabetic Nephropathy (FinnDiane) Study, long-term GV, defined by variability of glycated hemoglobin (HbA1c), was independently associated with progression of renal disease and cardiovascular events in patients with type 1 diabetes [4]. In the ADVANCE (Action in Diabetes and Vascular Disease: Preterax and Diamicron MR Controlled Evaluation) clinical trial visit-to-visit, GV was associated with both macro- and microvascular complications in type 2 diabetic subjects [5]. It has been shown that in patients with type 2 diabetes, increased GV predicts a decline in renal function [6], proliferative diabetic retinopathy and diabetic macular oedema [7], accelerated progression of coronary atherosclerosis [8], left ventricular adverse remodeling after myocardial infarction [9], major cardiovascular events, cardiovascular, and all-cause death [10]. The relationship between GV and mortality was described in the Verona Diabetes Study: in elderly type 2 diabetic subjects, the coefficient of variation of fasting plasma glucose turned out to be an independent predictor of total, cardiovascular, and cancer mortality [11]. In the trial comparing cardiovascular safety of insulin degludec vs. insulin glargine in patients with type 2 diabetes at high risk of cardiovascular events (DEVOTE), higher day-to-day fasting GV was associated with all-cause mortality and severe hypoglycemia [12]. The studies with continuous glucose monitoring recognized GV as a risk factor for hypoglycemia in type 1 and insulin-treated type 2 diabetic subjects [13,14].

At present, molecular mechanisms of deteriorating effect of high GV on the targeted organs are not fully understood. The data of experimental and clinical studies suggest that GV influence could be mediated through oxidative stress [15,16,17] and chronic low-grade inflammation [18,19]. It was shown that oscillating glucose produces more deleterious effects on endothelial function than constantly high glucose [15], and it enhances apoptosis of endothelial cells [16]. High glucose levels affect transcriptomic profile of many cells, including endothelial, mesangial, and pancreatic ones [20,21,22]. Many genes exhibit epigenetic changes under hyperglycemic condition [22]. It was demonstrated that transient high glucose induces persistent histone modifications, DNA methylation, changes in the number of 5-methylcytosines and 5-hydroxymethylcytosines in genomic DNA, and other modifications which alter the gene expression even after normalization of glucose level [23,24]. This mechanism is important for understanding the phenomenon of metabolic memory, which determines the long-term metabolic and pathogenic effects of hyperglycemia. Glucose-induced changes in mRNAs expression profiles are considered as another contributor to metabolic memory [20].

The complexity of the metabolic changes induced by glucose fluctuations, as well as a variety of glucose homeostasis regulators, is the rationale for an application of integrative network bioinformatics analysis to evaluate the molecular aspects of GV. The reconstruction and analysis of gene networks is a widely used approach for studying the mechanisms of complex biological processes and multifactorial diseases [25,26]. Specifically, this approach was applied for identification of key genes and pathways involved in pathogenesis of diabetes and its complications [27,28,29,30]. However, to our knowledge, GV-related networks have not been reconstructed and analyzed yet.

In recent years, artificial intelligence has provided new opportunities for data analysis in medical genomics [31]. Among them, the methods of automated retrieval of information from unstructured textual data (text mining) might be particularly helpful when complex processes involving genes, proteins, and phenotypes are considered [32]. In our study, we aimed to reconstruct and analyze gene networks related to GV in diabetes and its complications with the use of text-mining-based approach. For research purposes, we applied ANDSystem, which provides an automatic reconstruction and analysis of the networks based on the text mining of PubMed abstracts. The ANDSystem knowledge base contains about 30 million facts concerning genetic regulation, gene associations with diseases, protein-protein interactions, catalytic reactions, transport pathways, etc., extracted from more than 28 million PubMed abstracts [33,34].

## 2. Results and Discussion

### 2.1. Reconstruction and Analysis of the Networks Associated with Hyperglycemia and Hypoglycemia

At the first step, molecular networks related to hyperglycemia (Appendix A) and hypoglycemia (Appendix A) were reconstructed using the ANDSystem [33,34]. The gene network of hyperglycemia included 209 genes/proteins and 5818 interactions between them. The gene network of hypoglycemia included 128 genes/proteins and 2467 interactions. At the intersection of these gene networks, we identified 37 genes/proteins. These molecules were associated with both hyperglycemia and hypoglycemia, being highly connected with each other in the gene network (Appendix A, Figure 1). It can be assumed that these genes and proteins are most closely related to enhanced GV, which is characterized by the transition from hyperglycemia to hypoglycemia and vice versa. In what follows, we will call such genes as GV-related ones.

Ten GV-related genes showed the largest number of connections with other molecules in the network (Table 1). Among these molecules, as expected, there are insulin (*INS*) and glucagon (*GCG*), two principal regulators of blood glucose level in humans, as well as other hormones that affect glucose metabolism directly or indirectly, including leptin (*LEP*), insulin-like growth factor 1 (*IGF1*), somatostatin (*SST*), adiponectin (*ADIPOQ*), prolactin (*PRL*), ghrelin (*GHRL*), glucagon like peptide 1 receptor (*GLP1R*), proopiomelanocortin (*POMC*), amylin (*IAPP*), adrenomedullin (*ADM*), angiotensin II (*AGT*), secretin (*SCT*), and pancreatic polypeptide (*PPY*). The network includes genes encoding proinflammatory cytokines (*IL6, IL1B, TNF*), growth factors (*FGF2, IGF2*), molecules involved in the orchestration of insulin secretion and signaling (*GCK, KCNJ11, ABCC8, INSR, FLNA*), transcriptional factors (*HNF1A, FOS, ZGLP1*), neurotransmitter (*NPY*), enzymes (*DPP4, AKR1B1, ST3GAL4, MAP4K2, RAPGEF5*), and binding proteins (*ALB, IGFBP3, TAPBP).*

Thus, the network associated with GV includes a variety of molecular components. Hormones constitute a significant part of this network. This is consistent with the well-known fact that maintenance of glycemic fluctuations within the physiological limits is controlled by fine and well-balanced hormonal regulation. Circadian rhythms in secretion of growth hormone, melatonin, prolactin, cortisol, leptin, and ghrelin are highly correlated with sleep and can contribute to the intra-day glycemic fluctuations [35,36,37].

Among non-hormonal factors, interleukin-6 (*IL6*) showed the highest number of connections in the GV-related network. Circadian rhythm in IL-6 secretion with an elevation in day-time hours was described [38]. As IL-6 modulates glucose disposal and insulin sensitivity [39], one could speculate it can affect intraday GV. On the other hand, excessive glucose fluctuations can increase IL-6 production. In cultured human endothelial cells, an intermittent increase in glucose levels caused a more pronounced increase in *IL6* expression than a consistently high glucose [40]. The association between GV and plasma level of IL-6 has been shown in individuals with metabolic syndrome [41]. In patients with type 1 diabetes, an increase in the IL-6 levels was observed after a two-hour episode of hypoglycemia [42]. Hyperglycemia, following hypoglycemia, caused a further increase in IL-6 concentration [18].

The genes associated with insulin secretion and signaling turned out to be important nodes of the GV network. In humans, mutations in these genes (*GCK, HNF1A, ABCC8*, and *KCNJ11*) are causes of monogenic forms of diabetes known as maturity-onset diabetes of the young [43]. An increased GV was reported in patients with diabetes caused by mutations in the *HNF1A* [44] and *ABCC8* [45]. Mutations in the insulin receptor gene (*INSR*) are associated with insulin resistance and hyperglycaemia and can lead to hyperinsulinemic hypoglycaemia in adults and children [46,47]. The *GCK, KCNJ11, ABCC8*, and *INSR* were identified as genetic variants associated with type 2 diabetes and/or glycemic traits in genome-wide association studies [48]. However, the contribution of the polymorphisms of these genes to GV in healthy individuals and patients with diabetes has not been studied yet. The role of a number of molecules, identified as the network components, in GV phenomenon remains to be clarified.

At the next step, we performed the comparisons of expression patterns of GV-related genes in different tissues with the use of Bgee [49]. The greatest expression of GV-related genes was observed in the cardiovascular system, followed by pancreas, adipose and muscle tissues, gastrointestinal tract, and kidney; retina and nerves demonstrated a less maximal expression score (Appendix A). This fact gives further support to notion that GV could be the unifying link between impairment of insulin synthesis and signaling, glucose metabolism, and diabetic vascular complications.

Table 2 demonstrates the results of the Gene Ontology (GO) enrichment analysis by DAVID service [50], aimed at revealing the overrepresented biological processes for GV-related genes. In addition to the regulation of insulin secretion and different aspects of glucose metabolism, this analysis revealed some signaling pathways considered as the cornerstones in regulation of cellular metabolism (mitogen-activated protein kinase (MAPK) cascade, protein kinase B signaling, JAK-STAT cascade, regulation of peptidyl-tyrosine phosphorylation), cell cycle (regulation of mitotic nuclear division and cell proliferation), and cell-cell signaling. Positive regulation of nitric oxide biosynthetic process was identified among overrepresented processes.

It turned out that the biological processes identified as overrepresented are closely related to oxidative stress, glycation, endothelial dysfunction, and other processes that are considered as important players in the development of diabetes complications. In particular, oxidative stress can directly alter glucose homeostasis [51] and it itself is triggered by hyperglycemia [52]. The intensity of glycation is proportional to glucose concentration and depends on glucose metabolism [53,54]. MAPK cascade and nitric oxide biosynthesis are known as principal pathways in the biology of endothelial cells [55,56].

These data support the notion that GV-related genes could play an important role in pathogenesis of diabetic complications. We tested this assumption at the next stage of our work.

### 2.2. The Role of GV-Related Genes in the Networks of Diabetes Complications

The ANDSystem was used to construct gene networks associated with diabetic complications: cardiovascular disease (Appendix A), diabetic nephropathy (Appendix A), diabetic retinopathy (Appendix A), and diabetic neuropathy (Appendix A). For each of these gene networks, the role of GV-related genes was assessed by analyzing the betweenness centrality coefficients of the network participants (Table 3). Betweenness centrality reflects the number of shortest pathways between all pairs of genes in the analyzed gene network that go through a given gene. It describes the functional importance of a gene in a gene network.

Analysis shows that GV-related genes are significantly overrepresented in all networks of diabetes complications. In addition, the centrality of the GV-related genes was, on average, significantly higher than the average centrality of all participants of the considered gene networks (Table 3). These data support the key role of GV-related genes in the molecular orchestra of diabetes complications.

In order to find out what biological processes involved in pathogenesis of diabetic complications are triggered by glycemic fluctuations, the following analysis was carried out. Among the participants of the gene networks of diabetes complications, we selected those that were directly regulated by GV-related genes, according to ANDSystem. Thus, we identified 141 genes in the network of cardiovascular diseases, 219 genes in the network of diabetic nephropathy, 132 genes in the network of diabetic retinopathy, and 44 genes in the network of diabetic neuropathy. Further, using the DAVID service [50], the GO biological processes overrepresented for these genes were identified (Appendix A).

It was found that the response to hypoxia was overrepresented in all four sets of genes, and in all cases, it was among the top ten most significantly overrepresented biological processes. This suggests an important role of hypoxia as a mediator between GV and diabetes complications. Diabetes is characterized by altered oxygen deprivation signaling in the targeted organs, including the vasculature and kidney. Cellular response to hypoxia is controlled by hypoxia-inducible factor (HIF), a transcriptional regulator. The HIF family is considered as a master regulator of endothelial biology with rising interest in the field of diabetes-driven atherosclerosis [57]. It is postulated that a shift in the balance of HIF-1α and HIF-2α promotes proinflammatory and profibrotic pathways in glomerular and renal tubular cells [58]. A growing body of research indicates HIF-1α and its target genes as the contributors to the retinal neovascularization in diabetic retinopathy [59]. In obesity and type 2 diabetes, inflammation and hypoxia of adipose tissue may play an important role in the development of insulin resistance [60]. In its turn, intermittent hypoxia, which is used to improve the adaptation to hypoxia, reduced blood glucose in patients with type 2 diabetes [61]. These data are in agreement with our findings indicating the response to hypoxia as an important mechanism linking GV and chronic diabetic complications.

Other 15 biological processes were revealed to be overrepresented simultaneously for three sets of genes important for cardiovascular disease, diabetic nephropathy, and diabetic retinopathy (Table 4). They are involved in the inflammatory response, platelet function, angiogenesis, regulation of blood pressure, and nitric oxide biosynthesis. These processes, along with endothelial dysfunction, are considered as the cornerstones in the pathogenesis of diabetic vascular complications [62].

Molecular pathways, implicated in the regulation of cell cycle, proliferation, and cell-cell signaling, have been recognized among significantly overrepresented biological processes related to GV and complications. Among them, extracellular signal-regulated kinase (ERK) 1 and 2 cascade should be mentioned. ERK cascade plays a crucial role in multiple cellular processes, such as cell proliferation, differentiation, adhesion, migration, and survival [63]. Glucose can activate ERK1/2 in beta-cells, contributing to an increase in mitogenesis [64]. Moreover, activation of ERK is a component of insulin signaling. Accordingly, the targeting of the ERK cascade is considered as a potential treatment for insulin resistance and type 2 diabetes [65]. The activation of ERK1/2 is involved in pathogenesis of cardiac hypertrophy and dysfunction; recent studies showed the role of ERK in development of diabetic cardiomyopathy [66]. High glucose increases ERK phosphorylation in mesangial cells, promoting the synthesis of extracellular matrix, a key event in diabetic kidney disease [67].

Thus, it was revealed that GV-related genes occupy central positions in the gene networks of diabetes complications. The response to hypoxia, low-grade inflammation, abnormalities of angiogenesis and platelets function, disturbances in the regulation of blood pressure and vasodilatation, as well as alterations of cellular signaling pathways, could link GV with the development of diabetic complications.

### 2.3. Identification of New Candidate Genes in the GV Network

At the first step of this study, we identified by the ANDSystem the set of genes that are directly associated with both hyperglycemia and hypoglycemia, attributed them to GV. The Pathway Wizard tool of the ANDSystem allows one to identify indirect interactions between objects in the gene networks that pass through genes-intermediaries. Using this opportunity, it was possible to find genes/proteins for which there is no evidence on a direct connection with hypoglycemia and hyperglycemia, but there are indications of indirect relationships with glucose abnormalities through genes-intermediaries. As genes-intermediaries, we considered the GV-related genes described in the first section of this article. The genes/proteins that were associated with genes-intermediaries, we considered as candidate genes potentially involved in GV network. The analysis identified 334 such candidate genes. Further, in order to rank candidates and determine the most promising ones, gene prioritization was carried out using the ToppGene system [68]. The GV-related genes were used as a training set.

The results of prioritization could be found in Appendix A. The top ten most-priority genes are shown in Table 5. These genes are important for cell-cell interactions (*THBS1, FN1*), intracellular signal transmission (*MAPK1, STAT3, TP53*), stress response (*HSP90AA1, TP53, GSK3B, PTEN*), cell proliferation and differentiation (*MAPK1, TP53, EGFR, EGF*), apoptosis (*STAT3, TP53, GSK3B*), and glucose metabolism (*GSK3B*).

The *THBS1* gene received the highest priority. This gene encodes an adhesive glycoprotein thrombospondin 1 (THSP1), which regulates cell-cell and cell-matrix interactions. THSP1 is an important player in the regulation of adhesion, mobility, and growth of endothelial cells; it has important regulatory function in angiogenesis, inflammation, and tissue remodeling also [69]. Patients with diabetes and coronary artery disease have increased THSP1 levels in plasma [70]. The enhanced expression of *THBS1* was observed in mice with streptozotocin-induced diabetes. In cultured fibroblasts, high glucose increased expression of *THBS1* [71]. Being involved in atherosclerosis, inflammation, ischemia/reperfusion injury, cardiac hypertrophy, and heart failure, *THBS1* plays a significant role in the development of cardiovascular diseases [72]. In diabetic complications, this molecule is considered to be a key inductor of fibrosis [73].

Figure 2 shows the network of interactions between the *THBS1* and GV-related genes. It was demonstrated that *THBS1* is able to bind to CD36, a transporter and sensor of free fatty acids. Binding of CD36 to *THBS1* initiates anti-angiogenic signals, induction of endothelial cell apoptosis, and the inhibition of angiogenesis [74]. *THBS1* differentially regulates the release of IL-6 and IL-10 by human monocytes [75]. Both hypoglycemia and hyperglycemia increased serum concentrations of IL-6 [76,77], while hyperglycemia down-regulated *IL-10* mRNA levels [78]. *THBS1* modulates epidermal growth factor 1 (EGR1) function [79]. At the same time, up-regulation of *THBS1* is mediated by EGR1 [80]. Hyperglycemia is responsible for EGR1 activation, acetylation, and its prolonged expression [81].

Fibronectin is a component of the extracellular matrix, accumulating in the retinal and glomerular basement membranes and mesangial matrix in diabetic retinopathy and diabetic kidney disease [82,83]. High ambient glucose increases fibronectin production in human mesangial cells [84] and retinal pigment epithelial cells [85]. In mesangial cells, the effect was mediated via THSP and TGF-beta [84].

Heat shock protein 90α (Hsp90α), encoded by the *HSP90AA1* gene, is the stress-inducible isoform of the molecular chaperone Hsp90. The expression of *HSP90AA1* increases in some forms of cancer [86]. Previously, *HSP90AA1* was recognized by bioinformatic analysis as a node in a protein-protein interaction network in diabetic nephropathy [87]. However, the exact role of Hsp90α in diabetes and its complications remains to be clarified.

Both *EGF* and *EGFR*, encoding epidermal growth factor (EGF) and its receptor respectively, were present in the list of gene-intermediates. In diabetes, EGF/EGFR system was discussed primarily as a promoter of wound healing [88]. Recently it was demonstrated that EGF signal is involved in diabetes-induced vascular dysfunction, remodeling, and transcriptome dysregulation associated with renal involvement [89]. In patients with type 2 diabetes, soluble EGFR correlates with fasting blood glucose, insulin resistance, and glycated hemoglobin HbA1c [90]. Interestingly, EGFR can regulate the transcription of proopiomelanocortin [91], a hypothalamic appetite regulator and ACTH precursor modulated by glucose level [92]. Both EGFR and TP53 regulate expression of *IGFBP3* [93,94], which level is reduced in response to hyperglycemia [95].

Mitogen-activated protein kinase 1 (MAPK1), also known as ERK2, is a molecular target of metformin, a principal antihyperglycemic agent [96]. MAPK1 is one of the essential molecules implicated in canonical insulin signaling, an important modulator of protein synthesis [96]. In the liver, MAPK1 is involved in regulation of glucose and lipid metabolism required for physiological metabolic adaptation [97]. In mice on a high-fat/high-sucrose diet, MAPK1 knockout promoted insulin resistance and impairment of glucose tolerance [98].

Signal transducer and activator of transcription 3 (STAT3) is a principal modulator of β-cell survival and function [99]. It is involved in the regulation of hepatic insulin sensitivity and gluconeogenesis [100]. The excessive STAT3 signaling contributes to skeletal muscle insulin resistance [101]. Recent data indicated the role of STAT3 signaling in high-glucose-induced podocyte injury [102], tubular cell dysfunction [103], and diabetic retinopathy [104].

Tumor protein p53 (TP53), a tumor suppressor, was recently identified as a metabolic regulator. In the liver, TP53 improves insulin sensitivity via inhibition of MAPKs and NF-κB pathways [105]. The protein regulates the function of glucose transporters affecting their transcription and translocation. It negatively regulates glycolysis and positively regulates gluconeogenesis [106]. In the beta cells, TP53 mediates gluco- and lipotoxicity and induces apoptosis [106]. In patients with type 2 diabetes, the level of TP53 increased with the age, duration of diabetes, and waist-to-hip ratio [107].

Glycogen synthase kinase 3 beta (GSK3B) is a serine/threonine protein kinase that is involved in glycogen metabolism, regulation of gene transcription, cytoskeleton organization, cell cycle, and apoptosis. The enzyme inhibits glycogen synthase and decreases glycogen storage in skeletal muscles and, to a lesser extent, in the liver; it also deteriorates glucose tolerance and insulin sensitivity. Insulin inactivates GSK3B via protein kinase B (Akt) [108]. It was demonstrated that GSK3B modulates inflammatory response in the islets [109]. Women with gestational diabetes had increased GSK3B activity in skeletal muscle and adipose tissue, where the enzyme promoted the inflammatory response [110]. Blood glucose fluctuations suppressed the expression of phosphorylated GSK3B and contributed to cardiac fibrosis in a model of diabetic cardiomyopathy [111].

Phosphatase and tensin homolog (PTEN) is essential for both cellular growth and metabolism. PTEN counteracts insulin signaling via PI3K/Akt pathway and reduces insulin sensitivity in mouse adipose tissue, liver, and β-cells [112]. In humans, mutations in *PTEN* gene increase the risks of obesity and cancer, but decrease the risk of type 2 diabetes [113]. Recent experimental studies indicate the role of PTEN in β-cell function and apoptosis [99], hyperglycemia-induced angiogenesis suppression [114], and development of diabetic nephropathy [115].

As an indirect confirmation of the relevance of performed gene prioritization, we checked the co-occurrence of the candidate gene together with the terms denoting GV in full-text PubMed Central articles. We used information from full-text articles to test our prioritization because ANDSystem stores the information gathered from the abstracts, but not full-text articles. For each candidate gene, using the hypergeometric distribution, the likelihood of meeting the name of this gene, together with the terms denoting GV in full-text of PubMed Central articles, was estimated (Appendix A). At the next step, the Spearman’s rank correlation coefficient was calculated between the prioritization rank of a gene and the probability to discover the gene in articles together with the terms describing GV by chance. The correlation coefficient was found to be 0.3 and the statistical significance of the correlation was 1.6 × 10^−8^. This indirectly confirms the adequacy of the gene prioritization. It means that the top candidate genes are promising for experimental verification of their relationship with GV.

For further verification of obtained results, we used transcriptomic data stored in GEO (https://www.ncbi.nlm.nih.gov/geo/info/), the largest open repository of experiments in the field of gene expression. We were able to find only one study (GEO ID: GSE40779) related to glucose fluctuations in diabetes. In this study, carried out in *Rattus norvegicus*, non-coding RNA profile was assessed in the heart under glucose fluctuations compared to sustained hyperglycemia [116]. It was found that the expression of 20 of 350 scanned miRNAs significantly differed between diabetic hearts and diabetic hearts with glucose fluctuations [116]. We have conducted the search for the target genes of these 20 differentially expressed miRNAs in the miRTarBase database [117]. For 16 differentially expressed miRNAs, 69 target genes in *Rattus norvegicus* have been found. Four miRNAs (rno-miR-136, rno-miR-296*, rno-miR-326, and rno-miR-30c-2*) had no target genes in miRTarBase. At the next step, human orthologous genes were found for these 69 target genes using the NCBI Orthologs service (Appendix A). Next, we matched these genes with those we identified as the candidate genes most likely associated with GV.

It turned out that five genes, namely *PTEN, TNFRSF1A, CCND1, TLR2*, and *RELA*, which we proposed as the candidate genes for experimental verification, are targets of microRNAs differentially expressed in the GSE40779 experiment. According to the hypergeometric distribution, the probability of observing this for random reasons is low (*p*-value = 0.006). In addition, all of these five genes were among the top 100 candidate genes with highest priority predicted by us, which, according to the hypergeometric distribution, is unlikely by chance (*p*-value = 0.002). Moreover, three of five discussed genes were among the top 50 candidate genes with highest priority, which is also unlikely to be observed for random reasons (*p*-value = 0.03), and the *PTEN* was among the top ten most-priority candidates. This indirectly confirms the results of our bioinformatics analysis.

### 2.4. Study Limitations and Future Remarks

Our study is not without limitations. Taking into account the lack of the data on the molecular aspects of GV itself, we identified GV-related genes as those associated with both hyperglycemia and hypoglycemia. Consequently, some molecular aspects of physiological glucose fluctuations might not be recognized. Future reconstruction and analysis of tissue-specific gene networks may also be important. As the ANDSystem operates with PubMed data only, some relevant information could have been missed. Our study is a hypothesis-generating one. The role of newly identified candidate genes, as well as some molecular pathways and biological processes, needs further experimental verification.

## 3. Materials and Methods

Associative gene networks were built using the ANDSystem version: 20.0413b646 (ICG SB RAS, Novosibirsk, Russia) [33,34,118]. The computer tool ANDSystem was developed for automatic reconstruction of associative molecular-genetic networks describing the mechanisms of diseases and other complex phenotypic traits. It utilizes text mining, an automatic knowledge extraction from the texts of scientific publications. Previously, the ANDSystem was used to identify candidate genes associated with comorbidity of asthma and tuberculosis [119], preeclampsia, diabetes and obesity [120], primary open-angle glaucoma [121], asthma and hypertension [122], and lymphedema [123].

Comparative analysis of the expression of GV-related genes in the tissues was carried out by the “Expression comparison” function (https://bgee.org/?page=expression_comparison) provided on the website of Bgee resource [49].

Gene Ontology enrichment analysis was performed using the DAVID service version: 6.8 (LHRI, Frederick, MD, USA) [50] with the following parameters: organism—“*Homo sapiens*”, Gene_Ontology—“GOTERM_BP_DIRECT”. Gene Ontology biological processes were considered as overrepresented if the *p*-values with false discovery rate (FDR) correction were less than 0.05.

The calculation of betweenness centrality coefficients for the gene network participants was made by the Statistics function in the Analysis section of the ANDSystem [35].

The identification of candidate genes potentially involved in the regulation of GV was carried out using the Pathway Wizard tool of ANDSystem [118]. The parameter “Pathway length” was set equal to 4. “Object 1” was hypoglycemia, “Object 2”, “Object 3” and “Object 4”—any human gene/protein, “Object 5”—hyperglycemia. For the interactions “Object 1—“Object 2” and “Object 4”—“Object 5” the following “Interaction property” was set: “Direction”—any; all databases in the “Database filter” section; all types of interactions in the section “Interaction type filter”. For the interactions “Object 2”—“Object 3” the following “Interaction property” was set: “Direction”—“Right to left”; all databases in the “Database filter” section; all types of interactions in the section “Interaction type filter”, except “association”, “coexpression”, “expression”, “interaction”, “involvement”. For the interactions “Object 3”—“Object 4” the following “Interaction property” was set: “Direction”—“Left to right”; all databases in the “Database filter” section; all types of interactions in the “Interaction type filter” section, except for “association”, “coexpression”, “expression”, “interaction”, “involvement”. Genes presented at the “Object 3” were considered as the candidate genes.

The prioritization of candidate genes was carried out using the ToppGene system (BMI CCHMC, Cincinnati, OH, USA) [68]. A list of GV-related genes was used as a training set. The candidate gene list was used as a test set. The parameters in the “Training parameters” section have been set: “all Feature”.

The information about the function of the candidate gene products was obtained from the NCBI Gene database (https://www.ncbi.nlm.nih.gov/gene/).

The evaluation of the co-occurrence of the candidate gene together with the terms denoting GV in full-text PubMed Central articles was carried out as follows. All articles mentioning GV were found in PubMed Central interface (https://www.ncbi.nlm.nih.gov/pmc/) by the query: “glucose variability” OR “glycemic variability” OR “glycemia variability” OR “glucose fluctuations” OR “glycemic fluctuations” OR “sugar fluctuations” OR “glucose fluctuation” OR “glycemic fluctuation” OR “sugar fluctuation” OR “glucose excursions” OR “glycemic excursions” OR “unstable blood glucose” OR “fluctuating blood glucose”. For each candidate gene, all articles in PubMed Central mentioning this gene were counted. All articles in which the candidate gene name and a list of terms denoting GV occurred simultaneously were counted. Further, using these numbers, statistical significance of the co-occurrence of names of genes-candidates and GV was assessed. For this, a hypergeometric distribution, implemented in the hypergeom.sf function of the scipy.stats package [124] of the Python programming language, was used. The false discovery rate (FDR) correction (https://www.sdmproject.com/utilities/?show=FDR) for the multiple comparisons was carried out using the “Signed Differential Mapping” tool [125].

The correlation between the rank of a candidate gene and the probability of meeting the name of this gene together with terms denoting glycemic variability in full-text articles of PubMed Central was estimated using Spearman’s rank correlation coefficient. For this, the stats.spearmanr function of the scipy.stats package [124] of the Python programming language was utilized.

Search for Series of experiments of gene expression was performed in GEO (https://www.ncbi.nlm.nih.gov/geo/info/) by query: “a phrase denoting GV” AND (diabetic OR diabetes). The following phrases one by one have been used as “a phrase denoting GV”: “glucose variability”, “glycemic variability”, “glycemia variability”, “glucose fluctuations”, “glycemic fluctuations”, “sugar fluctuations”, “glucose fluctuation”, “glycemic fluctuation”, “sugar fluctuation”, “glucose excursions”, “glycaemia excursions”, “unstable blood glucose”, “fluctuating blood glucose”. The open experimental data used in our work [116] have the GEO ID: GSE40779 and are available by the link https://www.ncbi.nlm.nih.gov/geo/query/acc.cgi?acc=GSE40779. The target genes of differentially expressed miRNAs were extracted from the miRTarBase database [117]. Human genes that are orthologous of *Rattus norvegicus* genes, targeted by differentially expressed miRNAs, were found using the NCBI Orthologs service (https://www.ncbi.nlm.nih.gov/kis/info/how-are-orthologs-calculated/). Assessment of the statistical significance of the enrichment of candidate-gene list by genes, targeted by differentially expressed miRNAs, was performed using a hypergeometric distribution. For this purpose, we used the hypergeom.sf function of the scipy.stats package [124] of the Python programming language.

## 4. Conclusions

The identification of the molecular effects of GV in diabetes and its complications is a new challenge for Endocrinology and Molecular Medicine. In this study, we have performed for the first time the reconstruction and analysis of the GV-related network, using the bioinformatics analysis and text mining of publications indexed in PubMed. Firstly, we had constructed the network of GV, which included 37 genes/proteins related to both hyperglycemia and hypoglycemia. In this network, we have identified the genes of hormones regulating glucose metabolism, proinflammatory cytokines, growth factors, regulators of insulin secretion and signaling, transcriptional factors, and other molecules. The cardiovascular system, pancreas, adipose and muscle tissues, gastrointestinal tract, and kidney were recognized as the loci with the highest expression of GV-related genes. We have identified regulation of insulin secretion, glucose homeostasis, as well as some signaling pathways, which are involved in the regulation of cellular metabolism (MAPK cascade, protein kinase B signaling, JAK-STAT cascade, regulation of peptidyl-tyrosine phosphorylation), cell cycle (regulation of mitotic nuclear division and cell proliferation), and cell-cell signaling among biological processes that are most actively regulated by GV-related genes. We have shown that GV-related genes are central to the networks of diabetic complications, including cardiovascular disease, diabetic nephropathy, retinopathy, and neuropathy. Finally, we have identified new candidate genes (*THBS1, FN1, HSP90AA1, EGFR, MAPK1, STAT3, TP53, EGF, GSK3B*, and *PTEN*), promising for experimental verification of their role in the GV phenomenon. The obtained results provide further possibilities for the identification of new molecular markers and therapeutic targets in diabetes and its complications.

## Figures and Tables

**Figure 1 ijms-21-08691-f001:**
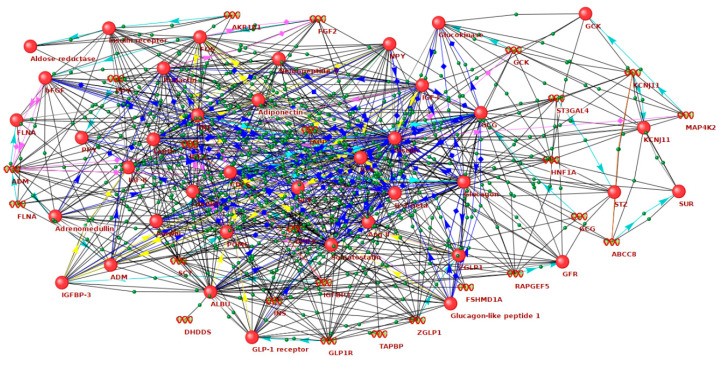
A network of the interactions of genes and proteins associated with both hyperglycemia and hypoglycemia (glucose variability (GV))-related genes), reconstructed by the ANDSystem.

**Figure 2 ijms-21-08691-f002:**
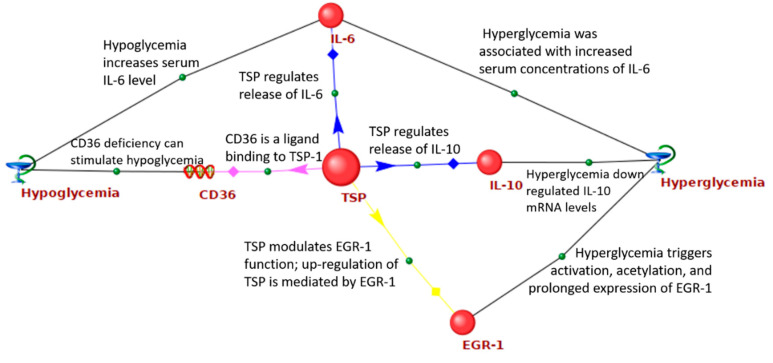
Regulatory network of interactions between *THBS1* and GV-related genes.

**Table 1 ijms-21-08691-t001:** The genes demonstrating the largest number of connections in the gene network of hypoglycemia and hyperglycemia.

Gene Symbol	Gene Product	Number of Connections
*INS*	insulin	59
*IL6*	interleukin 6	44
*ALB*	albumin	37
*GCG*	glucagon	33
*LEP*	leptin	30
*IGF1*	insulin-like growth factor 1	29
*SST*	somatostatin	29
*IL1B*	interleukin 1 beta	28
*ADIPOQ*	adiponectin, C1Q, and collagen domain containing	28
*PRL*	prolactin	27

**Table 2 ijms-21-08691-t002:** Gene Ontology biological processes overrepresented (*p*-values with false discovery rate (FDR) correction < 0.05) for a group of GV-related genes, identified with the DAVID service.

Gene Ontology Biological Processes	Gene Ontology ID	*p*-Values with FDR Correction
Regulation of insulin secretion	GO:0050796	1.38 × 10^−9^
Glucose homeostasis	GO:0042593	2.06 × 10^−6^
Positive regulation of MAPK cascade	GO:0043410	2.31 × 10^−5^
Positive regulation of protein kinase B signaling	GO:0051897	2.88 × 10^−5^
Positive regulation of glycogen biosynthetic process	GO:0045725	3.24 × 10^−5^
Positive regulation of mitotic nuclear division	GO:0045840	3.49 × 10^−4^
Glucose metabolic process	GO:0006006	3.92 × 10^−4^
Positive regulation of nitric oxide biosynthetic process	GO:0045429	3 × 10^−4^
Negative regulation of gluconeogenesis	GO:0045721	4 × 10^−4^
Positive regulation of cell proliferation	GO:0008284	2.8 × 10^−3^
Cellular protein metabolic process	GO:0044267	4.5 × 10^−3^
Positive regulation of smooth muscle cell proliferation	GO:0048661	6.2 × 10^−3^
Positive regulation of JAK-STAT cascade	GO:0046427	6.6 × 10^−3^
Cell-cell signaling	GO:0007267	1.08 × 10^−2^
Positive regulation of peptidyl-tyrosine phosphorylation	GO:0050731	1.9 × 10^−2^
Positive regulation of glucose import	GO:0046326	1.94 × 10^−2^

**Table 3 ijms-21-08691-t003:** Characterization of gene networks of the diabetes complications, taking into account the coefficients of betweenness centrality of their participants.

Parameter	Gene Network			
	Cardiovascular Disease	Diabetic Nephropathy	Diabetic Retinopathy	Diabetic Neuropathy
Number of participants	300	499	319	95
Number of interactions	4137	8252	4381	439
Number of genes associated with GV in the gene network	15	16	18	11
Statistical significance of the overrepresentation of genes associated with GV among all participants of the gene network	2.5 × 10^−14^	2.9 × 10^−12^	5.1 × 10^−18^	9.2 × 10^−15^
Average betweenness centrality coefficient for all participant of the network	361.54	595.32	362.02	98.5
Average betweenness centrality coefficient for the genes associated with GV	2764.76	4108.82	1958.24	397.94
Significance of difference between the coefficient of betweenness centrality of genes associated with GV and all genes in the network	3.9 × 10^−8^	2.7 × 10^−7^	1.3 × 10^−6^	1.8 × 10^−4^

**Table 4 ijms-21-08691-t004:** GO biological processes overrepresented (*p*-values with FDR correction < 0.05) for genes that are regulated by GV-related genes and participate in the gene networks of cardiovascular disease, diabetic nephropathy, and diabetic retinopathy.

Gene Ontology Biological Processes	Gene Ontology ID	*p*-Value of Overrepresentation in Cardiovascular Disease Network	*p*-Value of Overrepresentation in Diabetic Nephropathy Network	*p*-Value of Overrepresentation in Diabetic Retinopathy Network
Inflammatory response	GO:0006954	1.25 × 10^−9^	6.39 × 10^−16^	1.99 × 10^−7^
Regulation of blood pressure	GO:0008217	3.29 × 10^−10^	1.88 × 10^−5^	3.06 × 10^−6^
Positive regulation of angiogenesis	GO:0045766	7.09 × 10^−5^	5.60 × 10^−6^	2.25 × 10^−11^
Positive regulation of nitric oxide biosynthetic process	GO:0045429	1.13 × 10^−7^	1.23 × 10^−4^	2.18 × 10^−6^
Response to lipopolysaccharide	GO:0032496	2.15 × 10^−4^	7.50 × 10^−7^	8.35 × 10^−7^
Aging	GO:0007568	2.29 × 10^−4^	5.88 × 10^−12^	6.94 × 10^−8^
Positive regulation of ERK1 and ERK2 cascade	GO:0070374	4.15 × 10^−4^	2.24 × 10^−10^	1.86 × 10^−6^
Angiogenesis	GO:0001525	6.24 × 10^−4^	9.51 × 10^−6^	3.74 × 10^−6^
Response to drug	GO:0042493	2.7 × 10^−3^	5.30 × 10^−12^	4.23 × 10^−10^
Cell-cell signaling	GO:0007267	2.5 × 10^−3^	3.85 × 10^−4^	1.99 × 10^−5^
Positive regulation of cell proliferation	GO:0008284	3 × 10^−3^	7.56 × 10^−10^	6.29 × 10^−6^
Platelet degranulation	GO:0002576	4 × 10^−3^	3.42 × 10^−11^	1.29 × 10^−5^
Positive regulation of gene expression	GO:0010628	3.4 × 10^−3^	5.89 × 10^−4^	2.95 × 10^−5^
Positive regulation of phosphatidylinositol 3-kinase signaling	GO:0014068	1.9 × 10^−3^	2.67 × 10^−4^	1.94 × 10^−2^
Leukocyte migration	GO:0050900	1.41 × 10^−2^	1.11 × 10^−4^	8.6 × 10^−3^

**Table 5 ijms-21-08691-t005:** Top ten candidate genes potentially related to GV.

Rank	Gene Symbol	Gene Product	Gene product Characteristics According to NCBI Gene Database
1	*THBS1*	Thrombospondin 1	Glycoprotein, a component of extracellular matrix, which mediates intercellular interactions and plays a role in platelet aggregation, angiogenesis, and oncogenesis.
2	*FN1*	Fibronectin 1	Glycoprotein, a component of extracellular matrix, is involved in the cell adhesion, wound healing, blood coagulation, and tumor metastasis.
3	*HSP90AA1*	Heat shock protein 90 alpha family class A member 1	Chaperone promoting proper folding of target proteins during cell stress.
4	*EGFR*	Epidermal growth factor receptor	Receptor for members of the epidermal growth factor family. It is located on the cell surface and promotes cell proliferation.
5	*MAPK1*	Mitogen-activated protein kinase 1	MAP kinase that acts as a trigger for a variety of biochemical signals, being involved in cell proliferation, differentiation, and development.
6	*STAT3*	Signal transducer and activator of transcription 3	Signal transducer and activator of transcription in response to cytokines and growth factors. It mediates the expression of many genes in response to cellular stimuli and plays a key role in cell growth, apoptosis, and immune processes.
7	*TP53*	Tumor protein p53	Tumor suppressor protein that responds to various cellular stresses and regulates the expression of target genes. It causes a stop of the cell cycle, apoptosis, DNA repair, and metabolic changes. Mutations in this gene are associated with different human cancers.
8	*EGF*	Epidermal growth factor	Epidermal growth factor, which acts as a mitogenic factor by binding to a cell surface epidermal growth factor receptor. It plays an important role in the growth, proliferation, and differentiation of many cell types.
9	*GSK3B*	Glycogen synthase kinase 3 beta	Serine-threonine kinase, a member of the glycogen synthase kinase subfamily, regulates glucose homeostasis and is involved in energy metabolism, inflammation, mitochondrial dysfunction, and apoptosis.
10	*PTEN*	Phosphatase and tensin homolog	Phosphatidylinositol-3,4,5-triphosphate-3-phosphatase, a tumor suppressor. It inhibits AKT/PKB signaling pathway.

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
