# Peer review of "Bioinformatic Reconstruction and Analysis of Gene Networks Related to Glucose Variability in Diabetes and Its Complications"

_ijms, 2020, doi:10.3390/ijms21228691_

Round 1
Reviewer 1 Report
Dear authors,
the manuscript appears well written and the experimental design is good. However, as the authors explained yet in the Limitation section, a true experimental approach is lacking for validating the hypothesis.
Regard to the use of ANDSystem is quite innovative for the reconstruction of molecular maps on GV genes, but in my opinion this system should be used and should work together with proteomics/transcriptomics data processed by experimental cellular approaches.
The GV-related gene network is know... from the publications of experimental data, so this manuscript did not apport any advance in basic science.
So, the novelty of this work regards only bioinformatic validation of a pletora of data disseminated in literature.
I suggest to perform, in collaboration or not with the experts in the field a validation of the most significant networks involved in cardiovascular complications in a cellular model of GV.
Author Response
Author`s response to Reviewer 1
Dear Editors and Reviewer,
First of all, we would like to express our sincere appreciation for your positive feedback, valuable comments and suggestions. We have made every possible effort to improve the presentation of our study in accordance with the recommendations received. Below we provide a step-by-step response to the reviewer`s comments and suggestions.
Reviewer`s comment:
The GV-related gene network is know... from the publications of experimental data, so this manuscript did not apport any advance in basic science.
So, the novelty of this work regards only bioinformatic validation of a pletora of data disseminated in literature.
Response:
In this work we conducted a bioinformatic study of molecular mechanisms associated with glucose variability (GV) in diabetes. To our knowledge, this is the first study aimed to explore the molecular-genetic networks associated with GV, but not with diabetes itself. Indeed, we could not find any publications in PubMed for the query: ("gene networks" OR "gene network") AND ("glucose variability" OR "glycemic variability" OR "glycemia variability" OR "glucose fluctuations" OR "glycemic fluctuations" OR "sugar fluctuations" OR "glucose fluctuation" OR "glycemic fluctuation" OR "sugar fluctuation" OR "glucose excursions" OR "glycemic excursions" OR "unstable blood glucose" OR "fluctuating blood glucose"). The GV has only recently been identified as an independent risk factor for diabetic complications. The molecular mechanisms of increased GV in diabetes and its effect on target organs remain poorly understood. In this regard, it seemed important to us to analyze the GV phenomenon using bioinformatics approaches.
According to the reviewer`s comments, we have expanded the description of the relevance and the novelty of our work (please, see lines 35-36; 78-79).
Reviewer`s comment:
Regard to the use of ANDSystem is quite innovative for the reconstruction of molecular maps on GV genes, but in my opinion this system should be used and should work together with proteomics/transcriptomics data processed by experimental cellular approaches.
I suggest to perform, in collaboration or not with the experts in the field a validation of the most significant networks involved in cardiovascular complications in a cellular model of GV.
Author response:
We totally agree with the reviewer's proposal for experimental validation of our results. Unfortunately, at present there are no widely accepted approaches to the experimental modeling of glycemic variability both in vitro and in vivo. From our point of view, the development of these approaches deserves a separate study.
Nevertheless, according to the suggestion of Reviewer, we have performed additional search and included some open experimental data in our manuscript. Specifically, we searched for a Series of experiments in GEO (https://www.ncbi.nlm.nih.gov/geo/info/), the largest open repository of transcriptomic data of gene expression. The search was carried out by the query: "a phrase denoting GV" AND (diabetic OR diabetes). The following phrases were used one by one to denote GV: "glucose variability", "glycemic variability", "glycemia variability", "glucose fluctuations", "glycemic fluctuations", "sugar fluctuations", "glucose fluctuation", "glycemic fluctuation", "sugar fluctuation", "glucose excursions", "glycaemia excursions", "unstable blood glucose", "fluctuating blood glucose". We were able to find only one study (GEO ID: GSE40779) by the query: "glucose fluctuations" AND (diabetic OR diabetes). Nothing was found for other word combinations denoting GV.
The experiment GSE40779 performed a non-coding RNA profiling by array in the heart with glucose fluctuations compared with sustained hyperglycemia in Rattus norvegicus. It was found that the expression of 20 of 350 scanned miRNAs was significantly differed between diabetic hearts and diabetic hearts in condition of enhanced glucose fluctuations. We searched for target genes of these 20 differentially expressed miRNAs in the miRTarBase database (http://mirtarbase.cuhk.edu.cn/php/index.php). There were 69 of such target genes, known for Rattus norvegicus. At the next step, human orthologous genes for these 69 target genes were found using the NCBI Orthologs service (https://www.ncbi.nlm.nih.gov/kis/info/how-are-orthologs-calculated/). Next, we checked if among these 69 target genes are any candidate genes, which we proposed as potentially involved in GV at the section “2.3. Identification of new candidate genes in the GV network”.
It turned out that 5 genes PTEN, TNFRSF1A, CCND1, TLR2 and RELA, which we proposed as candidate genes for the experimental validation, are targets of microRNAs that were differentially expressed in the GSE40779 experiment. According to the hypergeometric distribution, the probability of observing this for random reasons is very low (p-value equal to 0.006). In addition, all of these 5 genes were in the top 100 candidate genes with the highest priority, which is also unlikely to obtain for random reasons (p-value = 0.002). Moreover, three of these 5 genes were in the top 50 candidate genes with the highest priority, which is also unlikely for random reasons (p-value = 0.03), and the PTEN gene was among the top 10 candidates with highest priority.
Thus, our results are consistent with the available experimental data on transcriptomic profiling of miRNAs in the Rattus norvegicus diabetic hearts with glucose fluctuations.
We have included these data in the manuscript. The additions could be found in lines 339-361; 424-438 of the text and in the Suppl. 12.
We have also changed the title of the article to “Bioinformatic reconstruction and analysis of gene networks related to glucose variability in diabetes and its complications” to emphasize that this work is theoretical.
The changes in the text of the manuscript are highlighted in blue.
We are hopeful that the changes made based on the reviewer`s comments improved the content of our manuscript and further increased the scientific value. Many thanks again for considering our work.
Yours sincerely,
Olga Saik and Vadim Klimontov
Reviewer 2 Report
Reviewer’s report
Title: Reconstruction and analysis of gene networks related to glucose variability in diabetes and its complications.
Authors: Olga Saik and Vadim Klimontov
General opinion:
Glucose variability (GV) is an important indicator of glycemic control and treatment efficacy in diabetic patients, correlating with the prevalence of complications and quality of life. However, the molecular mechanism underlying the link between GV and diabetic complications remain mostly unknown. In their work, Olga Saik and Vadim Klimontov applied bioinformatics tools to search for molecular targets and cellular pathways that could explain the association between GV and the course of diabetes. The analyses were carefully planned, correctly carried out from a methodological point of view and the manuscript is well written. Therefore I have only minor comments regarding the manuscript content and structure.
Minor revisions:
Introduction:
Lines 62-64: “Many genes exhibit epigenetic changes under hyperglycemic condition [22]. It was demonstrated that transient high glucose induces persistent histone modifications and alters gene expression even after normalization of glucose level [23, 24].” – It is worth to mention that histone modifications are not the only epigenetic changes related to hyperglycemia (these include, for instance, DNA methylation and changes in the number of 5-methylcytosines and 5-hydroxymethylcytosines in genomic DNA).
Results and Discussion:
2.1. Reconstruction and analysis of the networks associated with hyperglycemia and hypoglycemia
Lines 115-118 “Circadian rhythms in secretion of growth hormone, melatonin, cortisol, leptin, and ghrelin are highly correlated with sleep and can contribute to the intra-day glycemic fluctuations [36]." – prolactin is also secreted in circadian rhythm, and it can influence glucose metabolism [J Clin Psychopharmacol. 2006 Dec;26(6):629-31; Diabetes Care. 2013 Jul;36(7):1974-80.]
2.3. Identification of new candidate genes in the GV network
Lines 287-291 in the section regarding the mitogen-activated protein kinase 1 (MAPK1) it is also worth to mention that this is a main molecular target of metformin
2.4. Study limitations and future remarks
Lines 336-338: “The role of newly identified candidate genes, as well as molecular pathways and biological processes that have shown associations with GV, needs further experimental verification.” - This sentence is quite vague. The discussion shows that several identified metabolic targets and pathways have already been proven to be related to the course of diabetes and related complications. When writing about further directions, the authors should mention which pathways require validation in experimental research.
Author Response
Author`s response to Reviewer 2
Dear Editors and Reviewer,
First of all, we would like to express our sincere appreciation for your positive feedback, valuable comments and suggestions. We have made every possible effort to improve the presentation of our study in accordance with the recommendations received. Below we provide a step-by-step response to the reviewer`s comments and suggestions.
Reviewer`s comment:
Introduction:
Lines 62-64: “Many genes exhibit epigenetic changes under hyperglycemic condition [22]. It was demonstrated that transient high glucose induces persistent histone modifications and alters gene expression even after normalization of glucose level [23, 24].” – It is worth to mention that histone modifications are not the only epigenetic changes related to hyperglycemia (these include, for instance, DNA methylation and changes in the number of 5-methylcytosines and 5-hydroxymethylcytosines in genomic DNA).
Response:
Thank you very much. We have included the comment in the manuscript (please, see the lines 64-68).
Reviewer`s comment:
Results and Discussion:
2.1. Reconstruction and analysis of the networks associated with hyperglycemia and hypoglycemia
Lines 115-118 “Circadian rhythms in secretion of growth hormone, melatonin, cortisol, leptin, and ghrelin are highly correlated with sleep and can contribute to the intra-day glycemic fluctuations [36]." – prolactin is also secreted in circadian rhythm, and it can influence glucose metabolism [J Clin Psychopharmacol. 2006 Dec;26(6):629-31; Diabetes Care. 2013 Jul;36(7):1974-80.]
Response:
Thank you, we absolutely agree. We have included information on prolactin in the manuscript (lines 121-122).
Reviewer`s comment:
2.3. Identification of new candidate genes in the GV network
Lines 287-291 in the section regarding the mitogen-activated protein kinase 1 (MAPK1) it is also worth to mention that this is a main molecular target of metformin.
Response:
Thank you very much. We have added the notion that MAPK1 is a molecular target of metformin (lines 293-294).
Reviewer`s comment:
2.4. Study limitations and future remarks
Lines 336-338: “The role of newly identified candidate genes, as well as molecular pathways and biological processes that have shown associations with GV, needs further experimental verification.” - This sentence is quite vague. The discussion shows that several identified metabolic targets and pathways have already been proven to be related to the course of diabetes and related complications. When writing about further directions, the authors should mention which pathways require validation in experimental research.
Response:
We have performed additional search and included some open experimental data in our manuscript. The additions could be found in lines 339-361; 424-438 of the text and in the Supplement 12.
We have also changed the title of the article to “Bioinformatic reconstruction and analysis of gene networks related to glucose variability in diabetes and its complications” to emphasize that this work is theoretical.
We are hopeful that the changes we have made based on the reviewer`s comments improved the content of our manuscript and further increased the scientific value. Many thanks again for considering our work.
The changes in the text of the manuscript are highlighted.
Yours sincerely,
Olga Saik and Vadim Klimontov
Round 2
Reviewer 1 Report
Dear authors,
I appreciated the efforts of the authors in revising their paper in few days adding many other information that reinforce it.